# Biophysical Characterization of Adeno-Associated Virus Vectors Using Ion-Exchange Chromatography Coupled to Light Scattering Detectors

**DOI:** 10.3390/ijms232112715

**Published:** 2022-10-22

**Authors:** Christina Wagner, Bernd Innthaler, Martin Lemmerer, Robert Pletzenauer, Ruth Birner-Gruenberger

**Affiliations:** 1Analytical Development Europe, Takeda Vienna, 1220 Vienna, Austria; 2Gene Therapy Process Development, Takeda Orth an der Donau, 2304 Orth an der Donau, Austria; 3Institute of Chemical Technologies and Analytics, Technische Universität Wien, 1060 Vienna, Austria

**Keywords:** adeno-associated virus vectors, ion-exchange chromatography, multi-angle light scattering, dynamic light scattering, protein characterization, critical quality attributes

## Abstract

Ion-exchange chromatography coupled to light scattering detectors represents a fast and simple analytical method for the assessment of multiple critical quality attributes (CQA) in one single measurement. The determination of CQAs play a crucial role in Adeno-Associated Virus (AAV)-based gene therapies and their applications in humans. Today, several different analytical techniques, including size-exclusion chromatography (SEC), analytical ultracentrifugation (AUC), qPCR or ELISA, are commonly used to characterize the gene therapy product regarding capsid titer, packaging efficiency, vector genome integrity, aggregation content and other process-related impurities. However, no universal method for the simultaneous determination of multiple CQAs is currently available. Here, we present a novel robust ion-exchange chromatography method coupled to multi-angle light scattering detectors (IEC-MALS) for the comprehensive characterization of empty and filled AAVs concerning capsid titer, full-to-total ratio, absolute molar mass of the protein and nucleic acid, and the size and polydispersity without baseline-separation of both species prior to data analysis. We demonstrate that the developed IEC-MALS assay is applicable to different serotypes and can be used as an orthogonal method to other established analytical techniques.

## 1. Introduction

Recombinant adeno-associated virus (rAAV) vectors are currently the leading platform for delivering gene therapies in vivo for the treatment of severe and rare diseases in patients. rAAV vectors stand out by their low immunogenicity, long-term gene expression, non-pathogenic behavior and different tissue tropisms due to a vast variety of serotypes [1,2,3]. The genetic engineering of rAAV vectors that feature an improved transduction efficiency and cope with immunological barriers has been studied extensively in recent years [4]. To date, four gene therapeutics approved by the US Food and Drug Administration (FDA) or the European Medicines Agency (EMA) are on the market [5]. The first gene therapy product (Glybera), based on AAV1, was approved by the EMA in 2012 for the treatment of lipoprotein lipase deficiency, followed by Luxturna, an AAV2-based gene therapeutic, for the treatment of retinal dystrophy, which was approved by the FDA in 2017 and the EMA in 2018. The third, AAV9-based gene therapy product, Zolgensma, was approved by the FDA in 2019 for patients suffering from spinal muscular atrophy [1,6,7,8,9]. The fourth lentiviral-based gene therapeutic (Zynteglo) was approved by the EMA in 2019 and only recently by the FDA (September 2022) for the treatment of beta thalassaemia intermedia and major [10,11]. In addition, more than 200 gene therapeutics based on AAV vectors are currently investigated in clinical trials worldwide for the treatment of ocular diseases, cardiovascular diseases and cancer [12].

AAVs belong to the genus *Dependoparvovirus* within the family *Parvoviridae* [4]. They comprise non-enveloped, icosahedral capsids with the ability to insert single-stranded DNA up to 4.7 kb [5]. The capsid itself is made up from three viral proteins: VP1, VP2 and VP3. Those three viral proteins occur in a ratio of 1:1:10, with an overall sum of 60 interlocking proteins forming the icosahedral capsid structure [13,14]. The cellular tropism of the 12 AAV serotypes that have been identified so far is defined by differences in the receptor binding domains on the capsid surface arising from virion assembly [1,15]. The production of the VPs and capsid assembly are encoded by the *cap* gene—one of the open reading frames (ORF) located between two inverted terminal repeats, while the *rep* ORF is crucial for the replication and encapsidation of the viral genome [6,16].

To guarantee a safe and efficacious application of gene therapy products for patients, it is crucial to monitor the product quality to evaluate the critical quality attributes (CQAs), such as capsid titer, packaging efficiency (empty-to-full capsid ratio), viral genome integrity, aggregation content and other process-related impurities [17]. This demands robust and reliable analytical methods with a high throughput and little effort. To date, these attributes are assessed by different methods, varying in their precision and accuracy. While ELISA, PCR and light scattering are used to determine capsid titer, vector genome titer and aggregation content, respectively [18], analytical ultracentrifugation (AUC) and transmission electron microscopy (TEM) are still the leading platforms in the biopharmaceutical industry that provide insight into the quantity of empty, partially-filled and full capsids. Nevertheless, the long turnaround times and lack in high sample throughput limit their application in routine analysis and process development [17,19,20]. To circumvent these issues, orthogonal methods based on a chromatographic separation prior to sample analysis, such as size-exclusion chromatography (SEC) for the quantification of aggregates, can be used due to a faster response and a higher sample throughput. In combination with multi-angle light scattering (MALS), SEC provides a more detailed analysis of AAV samples, as the capsid titer, the polydispersity, the capsid size, the full-to-total capsid ratio, the absolute molar mass of the nucleic acid, the protein and the total capsid can all be assessed simultaneously without the need for column calibration [21,22]. However, the resolution of same-sized particles in the SEC columns is inherently impracticable due to the separation mechanism being based on the hydrodynamic radius of the particles. Furthermore, oligomeric forms are often not fully baseline separated from the monomers due to the properties of the SEC column. This can result in a deviation of the measured absolute molar mass of the monomer from the expected value, as the light scattering (LS) signals of the oligomeric forms are unproportionally higher than the ones of the monomer peak [21,23].

Ion-exchange chromatography (IEC) provides an alternative separation method, which is frequently used for the determination of empty and full AAV capsids. IEC allows more parameters to be optimized in order to enhance the chromatographic resolution of AAV populations compared to SEC, such as buffer medium, pH, temperature, flow rate, salt concentration and composition, gradient slope and column properties. The separation principle is based on the interaction of a positively charged (anion-exchanger) or negatively charged (cation-exchanger) stationary phase with complementarily charged AAV capsids [21]. The choice of ligands depends on the pH of the buffering systems and the stability of AAV vectors at that specific pH. When using an anion-exchange column, the pH of the buffer must exceed the isoelectric point (pI) of the AAV capsids to ensure an overall negative charge of the AAV particles and vice versa when applying a cation-exchanger [24,25,26].

Like SEC, IEC can be coupled to MALS and thereby enables a more comprehensive characterization of empty and filled AAV capsids due to a separation of both subpopulations prior to sample analysis. This novel application allows the determination of the capsid titer, the full-to-total capsid ratio, the polydispersity, the shape factor, the absolute molar mass of nucleic acid and protein, as well as the hydrodynamic radius and radius of gyration in one single measurement. Based on the physical principle of polarizability of matter, the intensity of scattered light (I_s_) at a certain angle (theta, θ) is directly proportional to the molecular weight (M), the concentration (c) of the analyte and the excess Rayleigh ratio (R) at angle theta assuming that the differential refractive index increment (dn/dc) is a constant value of 0.185 mL/g at ~660 nm (for unmodified proteins in aqueous medium) (Equation (1)) [22,27,28,29,30]:(1)$$\;I_{s}\left(\theta \right)\;\alpha \;c*M*{\left(\frac{\mathrm{dn}}{\mathrm{dc}}\right)}^{2}*R\left(\theta \right)$$

The weight-averaged molar mass can be determined as the intercept on the *y*-axis in the Zimm plot by extrapolating to angle zero (R(0°) = 1) and concentration zero (c = 0). The slope of the extrapolation of the concentration gives the z-averaged radius of gyration [31]. When analyzing AAVs with MALS, the molar masses of the protein and the nucleic acid can be measured simultaneously. This demands two detectors, e.g., UV/RI or a UV detector, measuring at two different wavelengths. In case of IEC, the AAV particles elute from the column by applying a linear salt gradient; thus, a RI detector becomes invalid due to the change in the refractive index caused by the increasing salt concentration introducing a bias in the data analysis. To circumvent this issue, dual wavelength UV-absorption detection can be used, where one wavelength monitors the nucleic acid content (absorption maximum at 260 nm), and the second wavelength detects the protein proportion (absorption maximum at 280 nm) [32].

In addition, the ratio of the absorbance at 260 and 280 nm (A260/280) allows the distinction of empty AAV capsids from filled ones. While values between 0.6–0.7 are indicative for empty AAVs, ratios between 1.3–1.4 represent filled capsids [32]. The ratios can be calculated by integrating the area of both peaks at 260 and 280 nm. A different approach was described by Porterfield et al., who developed a method to quickly assess the protein and nucleic acid content by using light scattering corrected UV absorbance spectroscopy and validated their results by comparison to the orthogonal SEC-MALS technology [33].

Because MALS measures the intensity of the scattered light, it not only allows the calculation of the absolute molar mass of the protein and nucleic acid but also the assessment of the geometric radius of the analyte(s). There is an angular dependency of the scattered light intensity and the particle size, more precisely, the radius of gyration (Rg) of a particle. The bigger the Rg, the greater the scattered light intensity at lower angles [29]. Consequently, it is possible to determine the size and size distribution of the individual components of the sample via the angular variation. However, there is a sensitivity limitation of the MALS detector. Particles with a Rg smaller than 10 nm show no angular dependency of the scattered light intensity. Light is scattered equally in each direction (= isotropic scattering). Compared to static light scattering (MALS), dynamic light scattering (DLS) provides a more sensitive approach and allows the determination of the hydrodynamic radius (Rh) of a particle down to a radius of 1 nm [29]. DLS is based on the time-dependent measurement of fluctuations in the intensity of the scattered light due to the Brownian motion of the particles, which is faster for smaller particles. Rh can then be calculated via the Stokes–Einstein relation (Equation (2)), with k_b_ being the Boltzmann constant, T the temperature, η the viscosity of the solution and D_t_ the diffusion coefficient of the particle [29]:(2)$$\;\mathrm{Rh}=\frac{k_{b}*T}{6{*\pi *\eta *D}_{t}}$$

Furthermore, the shape of a particle can be determined using the ratio of Rg and Rh. This is particularly important when analyzing heterogeneous samples comprising analytes of different shapes. Rg/Rh ratios of 0.77, 1 and >1 correspond to uniform spheres, hollow spheres and elongated particles, respectively [34].

Conclusively, IEC-MALS provides a detailed assessment of the biophysical properties of AAVs in heterogenous samples. Compared to SEC, where analytes are separated by their hydrodynamic radius, IEC is capable of chromatographically resolving analytes of the same size (Rh) but different overall charge e.g., empty and full AAV capsids. 

Here, we present a novel and robust IEC-MALS method for the characterization and quantification of empty and filled AAV capsids without the need for chromatographic baseline separation of both species prior to sample analysis. Our method provides excellent comparability with AUC and ELISA data. Sample recovery (R%) was between 70 and 100%, which is in line with other methods. Good linearity was obtained by diluting the sample to a capsid titer of 2.0 × 10^11^ cp mL^−1^ (CV < 5%). Furthermore, multiple sample injections yielded a high precision of the assay with a CV < 5% for capsid titer, hydrodynamic radius, polydispersity, full-to-total ratio, absolute molar mass of protein, nucleic acid and total capsid. In addition, the developed IEC method was applied to three different in-house produced serotypes (AAV5, AAV6 and AAV8) without the need for adapting the method conditions and validated by comparison to orthogonal methods, namely AUC and ELISA. 

## 2. Results and Discussion

For the development of an IEC-MALS method, AAV8 was the serotype of choice as it has been reported by Lock et al., to be separatable into filled and empty capsid populations using ion-exchange chromatography [35]. However, to determine whether an anion-exchange (AEX) or cation-exchange (CEX) column was needed, the isoelectric point of the selected serotype was measured using a capillary isoelectric focusing (cIEF) technique. Because the measured pI was 7.4 (net capsid), a CIMac AAV full/empty analytical column (anion-exchanger) was used for the development of the IEC assay. For sample binding and elution, buffers containing 20 mM Tris (pH 8.5, buffer B) and 20 mM Tris + 120 mM MgCl_2_ (pH 8.5, buffer E), respectively, were selected. The chromatographic separation was carried out using the gradient described in Appendix A. Due to the difference in their overall negative charge attributed to the encapsidated nucleic acid, empty AAV capsids elute earlier from the column than filled AAV capsids when increasing the salt concentration of the buffer. For sample detection, the AEX column was coupled to a UV detector, a static and a dynamic light scattering detector, which allowed the determination of the absolute molar masses of the protein and nucleic acid, the hydrodynamic radius, the radius of gyration and the polydispersity of the afore-separated empty and filled AAV capsid fractions. The capsid titer and the full-to-total ratio of the sample are additionally assessed. A schematic overview of an IEC-MALS method is given in Figure 1. Unlike SEC-MALS, which uses the UV absorption and the differential refractive index (dRI) detection for the calculation of the above-mentioned parameters, IEC-MALS demands dual wavelength UV-absorption detection, as an RI detector cannot be used when applying salt gradients due to a change in the refractive index with increasing salt concentration (dn/dc).

### 2.1. IEC-MALS Development

Like SEC-MALS, IEC-MALS requires the calibration and normalization of the MALS detector prior to sample analysis. Toluene was used as the standard for the calibration of the detector at 90°, while the remaining photodiode detectors were normalized to the 90° detector using bovine serum albumin (BSA), a monodisperse, isotropic scatterer [28]. Because the UV-Vis and light scattering detectors were operated in series, the resulting chromatograms showed shifts in the retention times as the sample is not detected simultaneously. As the sample progresses through the detectors, it becomes more diluted, and broader peaks are observed. To correct for these variations, an alignment and band broadening correction of the UV and LS signals were performed.

### 2.2. Determination of the UV Extinction Coefficients

The in the software integrated a “viral vector analysis” algorithm allows the calculation of the molecular weights of the total AAV capsid, the proportions of the protein and the transgene, provided that the UV extinction coefficients at 260 and 280 nm of the protein and nucleic acid are known. Because these parameters are specific for each serotype, we experimentally determined the UV extinction coefficients for AAV8 at both wavelengths using ASTRA 8.1 software. Therefore, two samples comprising mostly empty and mostly full AAV8 capsids were measured using an already established SEC-MALS method for the purification of AAV monomers from aggregates. Filled AAV capsids yielded UV extinction coefficients of 14.55 mL (mg cm)^−1^ and 24.65 mL (mg cm)^−1^ at 280 and 260 nm, respectively. Empty AAV capsids yielded UV extinction coefficients of 2.05 mL (mg cm)^−1^ and 1.39 mL (mg cm)^−1^ at 280 and 260 nm, respectively.

### 2.3. Comparison of %Filled AAV Capsids to Orthogonal Methods

Because AUC is used as the standard analytical technique for the quantification of empty and filled capsids as well as other AAV subspecies, results obtained by IEC-MALS were compared to AUC data regarding the full/empty (F/E) ratio [17,19]. Unlike IEC-MALS, AUC can resolve AAV capsids containing a partial genome from empty and full ones; however, it is a more time-consuming technique with low sample throughput. Another drawback is the need for large sample volumes and high capsid titers [17]. Prompted by this, we developed an IEC-MALS assay which provides a faster and simpler alternative for the determination of the F/E ratio with the advantage of receiving additional information (hydrodynamic radius, radius of gyration, polydispersity and absolute molar mass of protein and nucleic acid) about both AAV populations in one single measurement. Therefore, two AAV8 samples comprising mostly empty (meC) and mostly filled AAV capsids (mfC), respectively, were mixed at different ratios to obtain fractions of various F/E content ranging from 28% to 96% F/E (capsid titers: 1.0 × 10^13^ cp mL^−1^). In Figure 2, an excellent linear correlation between data obtained by IEC-MALS (measured %filled) and data generated by AUC (expected %filled) is observed, with a coefficient of determination (R^2^) of 0.9968, suggesting that IEC-MALS can be used alternatively for the determination of the F/E ratio. Because ion-exchange chromatography does not provide any information on subpopulations due to a lack of chromatographic resolution, data from AUC for partially-filled and filled particles were added up for the comparison with IEC-MALS data.

### 2.4. Linearity of the IEC-MALS Method

To test the sensitivity of the IEC-MALS assay, a sample containing an F/E ratio of ~62% was serially diluted covering a concentration range between 1.0 × 10^13^ cp mL^−1^ and 2.0 × 10^11^ cp mL^−1^. Good linearity was obtained when plotting the measured capsid titer against the expected capsid titer determined by ELISA with an R^2^ of 0.998 and a CV < 5% (Figure 3a). Sample recovery was between 70–100%. In addition, a linear correlation between the area of the UV signals and the expected capsid titer was observed with a R^2^ of 0.999 (Figure 3b). However, at lower sample concentrations, the coefficient of variation exceeded the 5% limit, probably due to the low sensitivity of the LS detector.

Since it is not possible to use neither UV detection at 230 nm (due to other absorbing components in the matrix at that specific wavelength) nor a fluorescence detector (because ASTRA software does not support this instrument in its configuration) as a second concentration detector for the calculations of the molar mass of the nucleic acid or the protein, the sensitivity of the method cannot be improved. Furthermore, ASTRA does not provide information on the signal-to-noise ratio; hence, the LOD and LOQ had to be assessed empirically. In addition, 8.3 × 10^10^ cp mL^−1^ was the lowest detectable analyte concentration, 2.0 × 10^11^ cp mL^−1^ was the lowest sample titer, which had been successfully quantified with a CV < 5% and a recovery of 73%. These results are in good agreement with the calculated LOD (8.3 × 10^10^ cp mL^−1^) and LOQ (2.5 × 10^11^ cp mL^−1^) using the data from Figure 2.

### 2.5. Robustness of the IEC-MALS Method

To check for the robustness of the optimized IEC-MALS method, different gradients and flow rates were tested with regard to F/E ratio, UV 260/280 ratio, absolute molar masses of the protein, and transgene. Figure 4 shows the obtained chromatograms for the varied linear gradients and flow rates, respectively. All chromatograms showed two distinct peaks corresponding to empty and filled AAV capsids. When changing the steepness of the salt gradient over the same time period from 0–35% buffer E to 0–45%, 0–55% and 0–65% buffer E using a flow rate of 0.5 mL min^−1^ (Appendix A), a shift in the retention times is observed. Regardless of whether a flatter ramp (0–35% buffer E) or a steeper gradient (0–65% buffer E) was applied, the calculated results do not differ significantly from one another but match expected values (Table 1). This proves that no baseline separation of empty and full AAV capsids is required prior to data analysis; hence, no further optimization of the developed IEC-MALS method was necessary. In SEC-MALS, however, the influence of the neighboring (aggregate) peak on calculated results of the monomer peak is greater due to the higher molar masses of multimer species impacting the data analysis of the monomers. When analyzing empty and filled AAV capsids, the light scattering intensity of both subspecies is the same; hence, IEC-MALS does not necessarily require baseline-separated empty/full peaks.

### 2.6. Application of the IEC-MALS Method to Other Serotpyes

Next, we applied the developed IEC-MALS assay to two different in-house produced serotypes, AAV5 and AAV6. Both serotypes were generated from different downstream process steps, to further prove that the IEC-MALS assay can be applied at different stages of the AAV manufacturing platform. In Figure 5, an overlay of the LS chromatograms and the “viral vector analysis” of AAV5, AAV6 and AAV8 is shown. Results were compared to orthogonal methods, such as AUC and ELISA with respect to full-to-empty ratio and capsid titer, respectively. For the comparison of the measured absolute molar mass of the protein to the expected one, the theoretical ratio (5:5:50, VP1:VP2:VP3) and molar masses (87 kDa (VP1), 73 kDa (VP2) and 62 kDa (VP3)) of the three virus protein subunits in an assembled AAV particle were used to calculate the expected molar mass of the protein [36]. The molar masses of the encapsidated transgenes of the different serotypes were calculated from the 5′-ITR to 3′-ITR of the respective plasmids using SnapGene software 5.1.5 (GSL Biotech LLC, Chicago, IL, USA).

For each serotype, two distinct peaks were generated corresponding to the empty and filled AAV populations. Best chromatographic separation of both species was obtained for serotype AAV8, which had been used for the development and optimization of the IEC-MALS assay. Despite the poor peak resolution of serotype AAV5, the measured F/E ratio of ~12% fits well to the expected F/E ratio of 10% (according to AUC). The molar mass of the protein was lower than the expected theoretical value of ~3.9 MDa; however, more accurate results would have been generated if the exact amino acid composition and the VP ratio of the assembled capsid had been known. This would require a more thorough investigation of the VP stoichiometry of each serotype prior to IEC-MALS analysis and will be investigated in continuing experiments but is beyond the scope of this paper. Because the proportion of the empty AAV capsids of serotype AAV6 was ~24% only (according to AUC), the CV of the measured hydrodynamic radius and radius of gyration was >5% due to a lack of sensitivity of the light scattering detectors. Similar results were obtained for the filled capsid fraction of serotype AAV5 (10% filled capsids) regarding the measured absolute molar mass of the nucleic acid, the hydrodynamic radius and the radius of gyration. For the remaining calculated parameters (capsid titer, full-to-total ratio, polydispersity, absolute molar masses of protein and encapsidated ssDNA), the CV was <5% (see Table 2).

### 2.7. Test of Different Weak and Strong AEX Columns

To evaluate the method performance, the IEC-MALS assay was tested with five different AEX columns of various providers. Comparable results between all AEX columns with respect to hydrodynamic radius, radius of gyration, capsid titer and absolute molar masses of protein and ssDNA were obtained (Figure 6). The CIMac AAv Full/Empty Analytical Column (BIA Separations, Ljubljana, Slovenia), the ProSwift SAX-1S Column (ThermoFisher Scientific, Waltham, MA, USA) and the UNO Q Polishing Column (BioRad, Hercules, CA, USA) are based on the strong basic nature of quaternary ammonium (QA) groups as counterions for the negatively charged AAVs and are therefore considered as strong anion-exchangers, while the ProSwift WAX-1S Column (ThermoFisher Scientific) and CIMac PrimaS Analytical Column (BIA Separations) are based on multimodal systems that coalesce anion-exchange chromatography and hydrogen bonding interactions. The performance of the UNO Q Polishing Column (BioRad) was diverging most within all five columns regarding the measured F/E ratio, capsid titer and radius of gyration. All other columns yielded comparable results regardless of whether a weak or strong AEX column had been used.

### 2.8. Salt Gradient vs. pH Gradient

Because the application of a pH gradient in combination with a strong AEX column leads to an increase in the interaction of the AAV particles with the QA ligands with rising pH due to the greater overall negative charge of the AAV capsids at high pH, a multimodal system (CIMac PrimaS Analytical Column) was selected for the chromatographic separation of empty and filled AAV capsids. Therefore, a linear gradient from pH 7 to 10 was used (Appendix A). As opposed to the pH gradient, the salt gradient yielded higher absolute molar masses of the protein (Table 3). This is probably because of the broad shape of the peaks obtained using the pH gradient (Figure 7), which impacts data analysis. An optimization of the pH gradient conditions, however, would have been out of the scope of this work.

## 3. Materials and Methods

### 3.1. Samples and Reagents

Tris (hydroxymethyl) aminomethane (Tris), magnesium chloride hexahydrate (MgCl_2_ × 6 H_2_O) and hydrochloric acid (HCl) were purchased from Merck (Darmstadt, Germany). Toluene, BSA and PBS were purchased from Sigma Aldrich (Saint Louis, MO, USA), Thermo Fisher Scientific (Waltham, MA, USA) and Aviva Systems Biology (San Diego, CA, USA), respectively. The binding buffer for the chromatographic separation of empty and filled AAV capsids was prepared by dissolving 20 mM Tris in ultra-purified water (Millipore Purification System, Merck, Darmstadt, Germany) and adjusting to pH 8.5 with 25% HCl. The binding buffer was prepared by dissolving 20 mM Tris and 120 mM MgCl_2_ in ultra-purified water and adjusting to pH 8.5 with 25% HCl. Both buffers were filtered through a 0.22 µm PES membrane (Steritop Millipore Express PLUS, Merck, Darmstadt, Germany).

The injection volumes of the samples ranged between 10 and 70 µL aiming a total injected mass of ~5–7 µg on the column.

### 3.2. Instrument Configuration

Analyses were performed on an Agilent HPLC 1260 system (Agilent, Waldbronn, Germany) equipped with a quaternary pump, a degasser, an autosampler and a UV-Vis detector monitoring the absorbance at 260 and 280 nm. In addition, the system was coupled online to a MALS detector (DAWN^®^, Wyatt Technology, Santa Barbara, CA, USA) with an integrated DynaPro^®^ NanoStar^®^ DLS detector (WyattQELS, Wyatt Technology, Santa Barbara, CA, USA). Prior to sample analysis, the calibration of the MALS detector at a scattering angle of 90° using toluene was carried out, and the remaining scattering angles were normalized to the photodetector at 90° using a monodisperse, isotropic scatterer. Here, we used a solution of 1% BSA (*w*/*v*) in PBS.

For chromatographic separation, CIMac™ AAV full/empty-0.1 Analytical Column was used and compared to four other anion-exchange columns provided by Bio-Rad Laboratories, Herculus, CA, USA (UNO Q Polishing Column), Thermo Fisher Scientific, Waltham, MA, USA (ProSwift™ SAX-1S Column and ProSwift™ WAX-1S Column) and BIA Separations, Ajdovscina, Slovenia (CIMac PrimaS™-0.1 Analytical Column).

Before analyzing the AAV samples, the anion-exchange columns were equilibrated according to the manufacturer’s instructions.

### 3.3. Data Processing

Unlike SEC-MALS, which operates with one eluent only, IEC-MALS requires a gradient for sample elution. Because the ASTRA 8.1 software (Wyatt Technology, Santa Barbara, CA, USA) is restricted to analyses based on one mobile phase only, OpenLAB CDS ChemStation A.02.02 (Agilent, Waldbronn, Germany) had to be additionally used. While the flow rate, the sample injections and the gradient settings were controlled via ChemStation, data analysis was performed with ASTRA 8.1 software solely. This software contains an algorithm (“viral vector analysis”), which is specifically designed for the analysis of virus vectors and provides information on the capsid titer, full-to-empty ratio and molar masses of the protein and encapsidated transgene by using specific input values attributed to the sample e.g., the molecule shape of the analyte (spherical), the extinction coefficients and dn/dc values of the protein and nucleic acid. By combining the sample parameters with light scattering technology and two concentration detectors targeting either the protein or nucleic acid content, the software calculates the above-mentioned sample characteristics using a series of equations, which are described in more detail by Wyatt [22].

## 4. Conclusions

In this work, we developed a robust and efficient IEC-MALS assay for the characterization and quantification of empty and filled AAV particles with respect to capsid titer, F/E ratio, polydispersity, hydrodynamic radius, absolute molar masses of the protein and the encapsidated transgene. We demonstrated that no baseline separation of the empty and filled AAV populations is required for calculation of the above-mentioned parameters. Furthermore, the IEC-MALS method is applicable to two other serotypes, AAV5 and AAV6, without the need for adapting method conditions. Results have shown good comparability with orthogonal methods, namely AUC and ELISA, regarding F/E ratio and capsid titer, respectively. Since IEC cannot resolve particles containing truncated versions of the transgene, the measured absolute molar mass of the encapsidated ssDNA can deviate from the expected theoretical value depending on the proportion of the less-filled AAV population in the sample. This should be taken into consideration when interpreting the data. Regardless of this, IEC-MALS provides an alternative analytical technique for the comprehensive characterization of AAV vectors with the advantage of covering a range of various critical quality attributes in one single measurement.

## Figures and Tables

**Figure 1 ijms-23-12715-f001:**
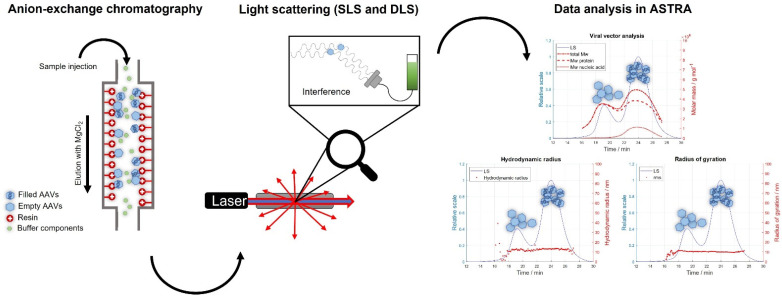
Schematic illustration of the IEC-MALS method. The AAV sample is loaded onto the anion-exchange column, eluted with a salt gradient containing MgCl_2_ and detected with multi-angle light scattering and UV detectors prior to data analysis using ASTRA software.

**Figure 2 ijms-23-12715-f002:**
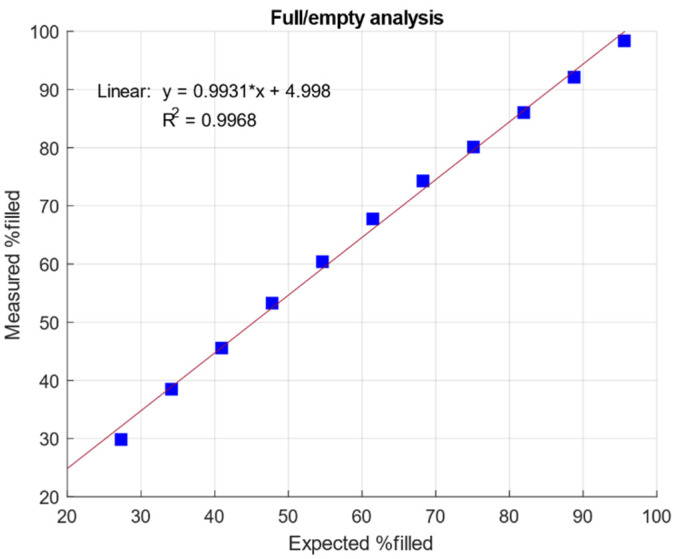
Linear correlation of measured % filled AAV capsids using IEC-MALS and expected % filled AAV capsids by AUC.

**Figure 3 ijms-23-12715-f003:**
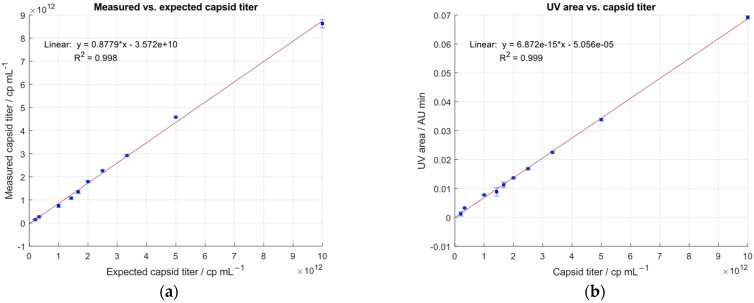
Linearity of the developed IEC-MALS method. Plots of expected capsid titers obtained by ELISA vs. (**a**) capsid titer measured with multi-angle light scattering and (**b**) UV area by integration of the UV profiles at 280 nm.

**Figure 4 ijms-23-12715-f004:**
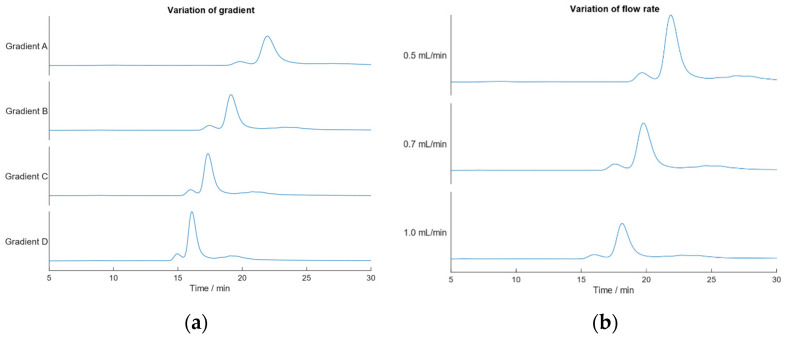
Evaluation of the robustness of the developed IEC-MALS method. (**a**) variation of the linear salt gradient. Gradient A: 0–35% buffer E, gradient B: 0–45% buffer E, gradient C: 0–55% buffer E and gradient D: 0–65% buffer E; (**b**) variation of the flow rate. Flow rates of 0.5 mL min^−1^, 0.7 mL min^−1^ and 1.0 mL min^−1^ were tested using a linear salt gradient from 0–35% buffer E.

**Figure 5 ijms-23-12715-f005:**
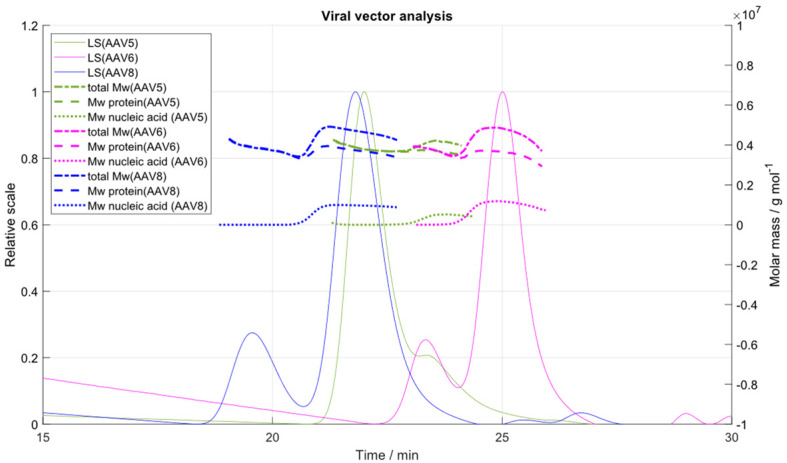
Overlay of the LS chromatograms (solid lines) and molar masses of the total capsid, protein and transgene (dashed lines) of serotypes AAV5 (green), AAV6 (pink) and AAV8 (blue).

**Figure 6 ijms-23-12715-f006:**
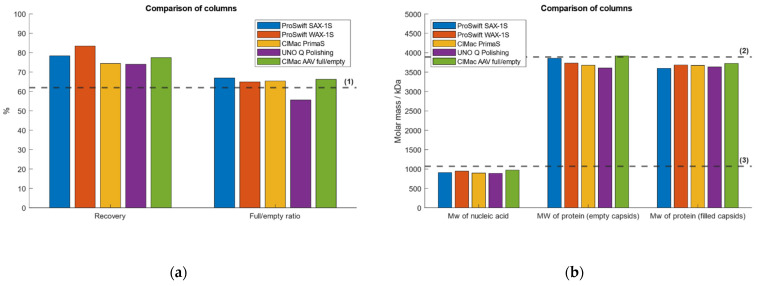
Evaluation of the method performance of the developed IEC-MALS assay using five different anion-exchange columns. The strong AEX columns ProSwift SAX-1S (blue), UNO Q Polishing (purple) and CIMac AAV full/empty (green) and the weak AEX columns ProSwift WAX-1S (orange) and CIMac PrimaS (yellow). Dashed lines (1)–(6) represent (**a**) the expected F/E ratio, (**b**) the absolute molar mass of the protein, the absolute molar mass of the encapsidated ssDNA, (**c**) the hydrodynamic radius, the radius of gyration of empty AAV capsids and the radius of gyration of filled AAV particles, respectively.

**Figure 7 ijms-23-12715-f007:**
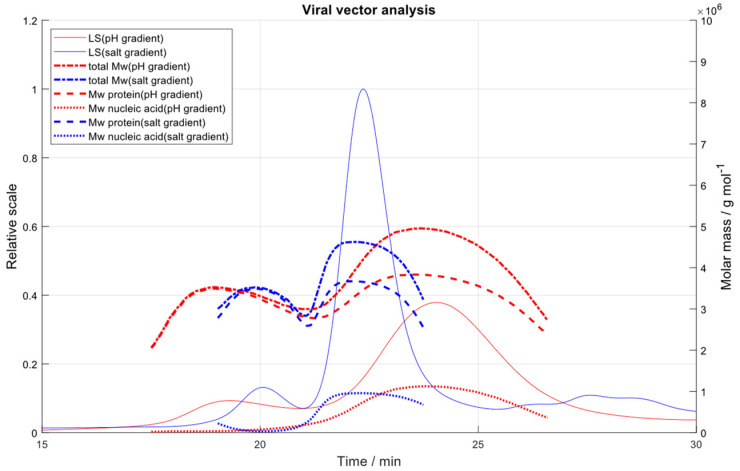
Overlay of the LS chromatograms and “viral vector analysis” of AAV8 obtained by a linear pH gradient (red) or linear salt gradient (blue).

**Table 1 ijms-23-12715-t001:** Overview of IEC-MALS method performance results of empty and full AAV8 capsids obtained by varying the gradient steepness and flow rate of the developed IEC-MALS method.

	Gradient A	Gradient B	Gradient C	Gradient D	Flow Rate 0.5 mL min^−^^1^	Flow Rate 0.7 mL min^−^^1^	Flow Rate 1.0 mL min^−^^1^
Capsid Titer ELISA/cp mL^−^^1^	1.00 × 10^13^	1.00 × 10^13^	1.00 × 10^13^	1.00 × 10^13^	1.00 × 10^13^	1.00 × 10^13^	1.00 × 10^13^
Measured Capsid Titer/cp mL^−^^1^	8.23 × 10^12^	8.50 × 10^12^	8.35 × 10^12^	8.40 × 10^12^	8.21 × 10^12^	8.75 × 10^12^	9.25 × 10^12^
RSD/%	1.5	0.3	0.6	0.8	4.3	0.8	0.3
Recovery/%	82	85	84	84	82	88	92
Expected %full	62	62	62	62	62	62	62
Measured %full	66	66	66	67	65	59	59
Difference %full/%	6	6	7	7	5	4	5
Expected Mw Nucleic Acid/kDa	1070	1070	1070	1070	1070	1070	1070
Measured Mw Nucleic Acid/kDa	959	949	956	951	936	904	902
RSD/%	0.1	0.7	0.8	0.7	1.9	0.9	0.1
Difference MW Nucleic Acid/%	10	11	11	11	13	16	16
Measured Mw Protein (Empty)/kDa	3842	3798	3809	3819	3750	3652	3573
RSD/%	1.7	0.5	0.5	0.5	1.0	0.8	0.4
Measured Mw Protein (Full)/kDa	3713	3690	3694	3695	3657	3659	3655
RSD/%	0.3	0.3	0.4	0.3	0.1	0.3	0.0
UV 260/280 Ratio (Empty)	0.69	0.68	0.68	0.65	0.66	0.46	0.74
UV 260/280 Ratio (Full)	1.34	1.32	1.32	1.33	1.34	1.31	1.33

**Table 2 ijms-23-12715-t002:** Overview of IEC-MALS method performance results of empty and filled AAV8 capsids of different serotypes.

	AAV5	AAV6	AAV8
Capsid Titer ELISA/cp mL^−^^1^	6.73 × 10^13^	2.12 × 10^12^	1.00 × 10^13^
Measured Capsid Titer/cp mL^−1^	6.20 × 10^13^	2.28 × 10^12^	7.74 × 10^12^
RSD/%	3.3	1.4	2.1
Recovery/%	92	107	77
Expected %full	10	68	62
Measured %full	12	66	66
Difference %full/%	15	3	7
Expected Mw Nucleic Acid/kDa	760	1240	1071
Measured Mw Nucleic Acid/kDa	457	1128	970
RSD/%	4.9	1.6	0.9
Difference MW Nucleic Acid/%	40	9	9
Measured Mw Protein (Empty)/kDa	3759	3850	3920
RSD/%	0.8	0.9	1.3
Measured Mw Protein (Full)/kDa	3466	3627	3724
RSD/%	3.1	0.3	0.5
rh (Empty)/nm	12.5	8.4	13
RSD/%	0.3	13.5	2.3
rh (Full)/nm	9	12	14
RSD/%	10.4	0.9	0.5
rms (Empty)/nm	9	8	9
RSD/%	22.4	35.2	11.8
rms (Full)/nm	9	8	8
RSD/%	6.8	13.1	16.4
Polydispersity	1.011	1.040	1.012
RSD/%	1.0	0.3	0.2

**Table 3 ijms-23-12715-t003:** Comparison of IEC-MALS method performance results of empty and filled AAV8 capsids obtained by a linear pH gradient and a linear salt gradient.

	pH Gradient	Salt Gradient
Capsid Titer ELISA/cp mL^−^^1^	1.00 × 10^13^	1.00 × 10^13^
Measured Capsid Titer/cp mL^−^^1^	9.11 × 10^12^	8.87 × 10^12^
RSD/%	0.8	1.5
Recovery/%	91	89
Expected %full	62	62
Measured %full	58	68
Difference %full/%	6	9
Expected Mw Nucleic Acid/kDa	1070	1070
Measured Mw Nucleic Acid/kDa	1014	986
RSD/%	0.6	1.1
Difference MW Nucleic Acid/%	5	9
Measured Mw Protein (Empty)/kDa	3359	3802
RSD/%	0.7	1.1
Measured Mw Protein (Full)/kDa	3670	3818
RSD/%	0.3	0.3
UV 260/280 Ratio (Empty)	0.77	0.83
UV 260/280 Ratio (Full)	1.26	1.34

## Data Availability

Not applicable.

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
