# Peer review of "Biophysical Characterization of Adeno-Associated Virus Vectors Using Ion-Exchange Chromatography Coupled to Light Scattering Detectors"

_ijms, 2022, doi:10.3390/ijms232112715_

Round 1
Reviewer 1 Report
This work describes a well-executed application of IEC with light scattering and UV detection to characterize AAV. The results indicate the approach works. The writing is mostly very clear, except the one point in major comments below.
Major comments -
The details of the IEX method - the flow rates, gradient details, buffer A., B, C, D, E...gradient A, B, C, D, and the pH gradient details do not seem to all be presented. I could not find a clear description of those details. These should be clearly included in the method section, or at least the supplemental methods section (a table would seem most clear) otherwise your work may not be replicated.
Minor comments -
A little more detail on how the (“viral vector analysis”) works will be helpful - this could be added to supplementary materials, or referenced if there is a citable reference for this.
line 61: although the separation concept plus light scattering and UV/RI is more sophisticated there is value in a rapid estimate of titer and %full using UV spectroscopy with detection at 260 and 280 nm for a whole sample without separation. I think you should reference Porterfield where 2 unknowns (protein and DNA) can be determined by 2 measurements (A260 and A280) - this method can be adapted to convert results to the equivalent total capsids and total DNA, giving titer and % full - see Porterfield JZ, Zlotnick A 2010. A Simple and General Method for Determining the Protein and Nucleic Acid Content of Viruses by UV Absorbance. Virology 407(2):281-288. - this is similar to what you are doing with the A260 and A280 but not exactly the same.
Author Response
thank you for your review!
ad comment "line 61: although the separation concept plus light scattering ...."
I cited Porterfield in line 120 because I think it fits better in this section
for the remaining comments please see the attachment

Reviewer 2 Report
Wagner and colleagues are presenting an IEC-MALS method that can be used for multi attribute monitoring of gene therapy vectors, namely recombinant AAVs. Their workflow allows for characterization of AAV samples regarding titre, fill state, absolute mass of the protein and cargo genome as well as size and polydispersity. Additionally, they have tested their method using multiple commonly used serotypes, AAV5, 6 and 8 and it was validated via AUC and ELISA. Their workflow can certainly be considered highly interesting for the field. Furthermore, the manuscript is well written and figures nicely illustrate the author’s results.
My two main concerns are regarding the intact capsid mass and the lack of baseline separation using IEC. I recommend further validation and optimization to address those two points. Please see below for more detailed comments:
- Introduction: The introduction is very comprehensive, providing valuable background information about AAVs and MALS. Zynteglo was approved in September, I recommend adding this to the list.
- Results and discussion:
o Why do the authors claim that E/F AAV9 cannot be separated using IEC? Please comment.
o Can the authors please specify how much sample volume and capsid titre is required for their method? I recommend using data shown in Fig 2 to calculate LOD and LOQ.
o Please make sure abbreviations are explained upon first use. E.g. cp/ml?
o Fluorescence detection would offer greater levels of sensitivity. Can the authors please comment on why it’s not possible to use FLR detection?
o I have some concerns about the theoretical capsid mass calculated based on the expected VP ratio of 5:5:50. As recently shown by Albert Heck and co-workers (https://pubmed.ncbi.nlm.nih.gov/33712599/) capsid stoichiometry can differ significantly, especially using in house produced samples. Can the authors validate the VP ratio and corresponding capsid mass using e.g. another chromatographic method (RP or HILIC)? Thereby they might be able to explain observed differences in mass (for example mentioned in line 297 onwards).
o Table 2: Shown are the intact mass in kDa of empty and full capsids. Please explain the difference between the two of them. This results in my opinion again shows the needs for validation.
o Another point is the lack of baseline separation between full and empty capsids. I understand that baseline sep is apparently not necessarily needed using IEC-MALS but the authors claim that IEC offers great potential for optimization. They seem to have tested many different conditions, so I’m wondering why separation was not achievable in this case while other studies in literature using IEC succeeded. Please comment on this.
Author Response
Thank you for your review!
ad comment "Introduction: The introduction is very comprehensive, providing valuable background information about AAVs and MALS. Zynteglo was approved in September, I recommend adding this to the list." -done
ad comment "Results and discussion:"
o Why do the authors claim that E/F AAV9 cannot be separated using IEC? Please comment.
We have tried separating AAV9 on AEX and CEX columns, however, we were not able to do so and we are not quite sure why. However, we are still investigating this issue. I have removed this sentence, just to be on the save side and not to raise any questions that we cannot answer (yet).
o Can the authors please specify how much sample volume and capsid titre is required for their method? I recommend using data shown in Fig 2 to calculate LOD and LOQ. - please see the attachment
o Please make sure abbreviations are explained upon first use. E.g. cp/ml? -done;
o Fluorescence detection would offer greater levels of sensitivity. Can the authors please comment on why it’s not possible to use FLR detection? - please see attachment
o I have some concerns about the theoretical capsid mass calculated based on the expected VP ratio of 5:5:50. As recently shown by Albert Heck and co-workers (https://pubmed.ncbi.nlm.nih.gov/33712599/) capsid stoichiometry can differ significantly, especially using in house produced samples. Can the authors validate the VP ratio and corresponding capsid mass using e.g. another chromatographic method (RP or HILIC)? Thereby they might be able to explain observed differences in mass (for example mentioned in line 297 onwards). - please see attachment, line 306 onwards
o Table 2: Shown are the intact mass in kDa of empty and full capsids. Please explain the difference between the two of them. This results in my opinion again shows the needs for validation. – it always depends on the sample concentration; due to lower capsid titer, we could not apply as much sample as we wanted, hence, the noise might be higher, which results in a slight variation of the calculated Mw from the actual Mw.
AAV5 contains 10% filled capsids only, so there is a higher noise resulting in a higher RSD. In addition, we used an automatic data analysis method, which we applied to all other samples, if we did it manually, then the values of empty and filled would probably match better.
o Another point is the lack of baseline separation between full and empty capsids. I understand that baseline sep is apparently not necessarily needed using IEC-MALS but the authors claim that IEC offers great potential for optimization. They seem to have tested many different conditions, so I’m wondering why separation was not achievable in this case while other studies in literature using IEC succeeded. Please comment on this. – we have tried several gradient conditions as well and tested other studies on our instruments, however, we were not able to easily separate both peaks as these studies suggested. However, we saw, that it was not necessary to achieve a baseline separation of both peaks, which makes it easier to apply the developed method to multiple serotypes without the need of adaption.
